# Characteristics associated with women undergoing their first mammography screening at a younger age

Eiman Alawadhi[1], Alyah A. Al-Yahya[2¶], Batool A. Mohammed[2¶], Mariam M. Al-Ruwaie[2¶], Sarah H. Al-Mutairi[2¶], Nourah M. Alsheridah[3]*

1 Department of Epidemiology and Biostatistics, College of Public Health, Kuwait University, Kuwait,
2 Department of Public Health, Ministry of Health, Kuwait, 3 Inequalities in Cancer Outcomes Network, London School of Hygiene and Tropical Medicine, London, United Kingdom

¶ These authors contributed equally to this work.
* nourah.alsheridah@lshtm.ac.uk

## Abstract

Breast cancer is the most commonly diagnosed cancer and a leading cause of cancer death among women. Kuwait has among the highest age-standardized incidence in the Gulf, yet mammography uptake remains low. The 2024 USPSTF recommends routine screening from age 40 for average-risk women. To identify characteristics associated with undergoing first-time mammography at ages 40–49 versus ≥50 in Kuwait. Cross-sectional analysis of 5,242 asymptomatic Kuwaiti women aged ≥40 with no prior screening in the Kuwait National Mammography Screening Program, 2014–2019. Multivariable logistic regression estimated adjusted odds ratios (aORs) and 95% CIs. Overall, 65.0% had their first screen at 40–49. Younger screening increased over time (per-year aOR 1.10, 95% CI 1.06–1.14; Cuzick z = 4.56, p < 0.001). Higher odds were observed with university or postgraduate education versus illiterate to secondary (aOR 4.53, 3.71–5.53; p < 0.001), parity 3–5 versus none (2.66, 1.85–3.82; p < 0.001), clinical breast examination (1.39, 1.11–1.74; p = 0.004), hormone replacement therapy use (1.69, 1.28–2.23; p < 0.001), and citing social media as the main information source (2.58, 1.28–5.19; p = 0.008; n = 94, 1.8%). Lower odds were seen for overweight (0.67, 0.53–0.85; p = 0.001) and obesity (0.53, 0.42–0.66; p < 0.001) versus normal BMI, hysterectomy (0.51, 0.34–0.76; p = 0.001), and longer breastfeeding (≥24 months versus never to <1 month: 0.41, 0.30–0.57; p < 0.001). Earlier first screening was associated with higher education, greater parity, clinical contact, hormone therapy use, and digital information sources, and was less likely with higher BMI, hysterectomy, and prolonged breastfeeding. Targeted, culturally tailored outreach to women with lower education or higher BMI, along with evaluated digital strategies, may promote earlier participation. Prospective follow-up is needed to determine whether earlier first screening leads to down-staging and improved outcomes.

**Data availability statement:** The data underlying this study are owned by the Ministry of Health – Kuwait and cannot be deposited in a public repository due to national regulations and ethical restrictions. Qualified researchers may request access from the Standing Committee for Coordination of Medical Research at the Health Information Office, Ministry of Health, Kuwait (Ground Floor; contact: hkhamis@moh.gov.kw). Access will be granted following review and approval by the Ministry.

**Funding:** The authors received no specific funding for this work.

**Competing interests:** The authors have declared that no competing interests exist.

## Introduction

Breast cancer is the most commonly diagnosed cancer and a leading cause of cancer-related death among women globally, accounting for 11.7% of all cancer cases and 6.9% of cancer deaths [1]. Incidence rates are particularly high in high-income countries, partly due to reproductive and lifestyle risk factors (e.g., delayed childbearing, lower parity, obesity) and greater uptake of screening [2]. Although advances in treatment have improved outcomes, early detection remains a cornerstone for improving survival [3]. Mammography is the most effective method for detecting breast cancer at an early stage, often before symptoms appear, enabling earlier treatment and better prognosis [4]. While international guidelines such as those from the U.S. Preventive Services Task Force (USPSTF) traditionally recommended routine mammography beginning at age 50, the 2024 update lowered this to age 40 for women at average risk [5]. In Kuwait and across the Gulf region, national screening programs also target women beginning at age 40, aligning with this shift.

In Kuwait, breast cancer is the most frequently diagnosed cancer among women [6]. The Kuwait National Mammography Screening Program (KNMSP), the country's sole population-based screening program, began in 2014 to offer population-based screening for asymptomatic Kuwaiti women aged 40–69 [7]. Uptake remains low (7.8% screened via KNMSP in its first five years vs a 70% target), and private-sector mammography, largely used by non-Kuwaiti residents, lies outside KNMSP and is not centrally recorded.

Given the persistently low screening rates and rising incidence of breast cancer, it is essential to understand the factors that influence women's decisions to undergo mammography—especially among women aged 40–49, who are already targeted within the national screening program [7]. Identifying the characteristics of those who undergo screening at a younger age can support the development of more tailored and effective interventions to improve participation and promote early detection.

Although prior research has explored predictors of breast cancer screening participation globally, few studies have examined the specific characteristics of women undergoing their first mammogram at a younger age [8]. Studies from various countries have consistently shown that factors such as education, socioeconomic status, and healthcare access significantly influence participation in breast cancer screening programs [9,10], but these findings may not directly apply to Kuwait's unique healthcare and cultural context. This study aims to describe the characteristics of women undergoing their first mammography screening at the KNMSP and to assess the factors associated with undergoing screening at age 40–49 compared to age 50 and older.

## Materials and methods

### Study design and participants

This cross-sectional study used secondary KNMSP data to describe characteristics of women undergoing their first mammography screening and to assess factors associated with screening at 40–49 vs 50+years. Data covered screenings from April 1, 2014 to March 31, 2019. Eligibility was limited to asymptomatic women attending

routine KNMSP screening for a first mammogram; diagnostic referrals were not eligible. Symptom status was self-reported. KNMSP enrolls Kuwaiti nationals; private-sector mammography is outside KNMSP and is not systematically captured.

Of the 19,984 Kuwaiti women screened during this period, exclusions were applied sequentially to derive the analytic cohort. First, duplicate records were removed to retain only first-time screens (n = 10,902 excluded; 9,082 remaining). Next, women who reported any prior mammogram outside KNMSP were excluded (n = 3,784; 5,298 remaining), as were those with missing prior mammography history (n = 48; 5,250 remaining). Finally, women younger than 40 years at screening were excluded (n = 8), yielding a final sample of 5,242 asymptomatic Kuwaiti women aged ≥40 years undergoing their first KNMSP mammogram. Participants were drawn from all six governorates, reflecting the geographic distribution of screening centers. A flow diagram of inclusion and exclusion is presented in Fig 1.

### Data collection and procedures

Data were collected by trained nurses at the KNMSP. Prior to undergoing screening, each participant completed a structured questionnaire, and responses were directly entered into an electronic database by the nurses and radiology

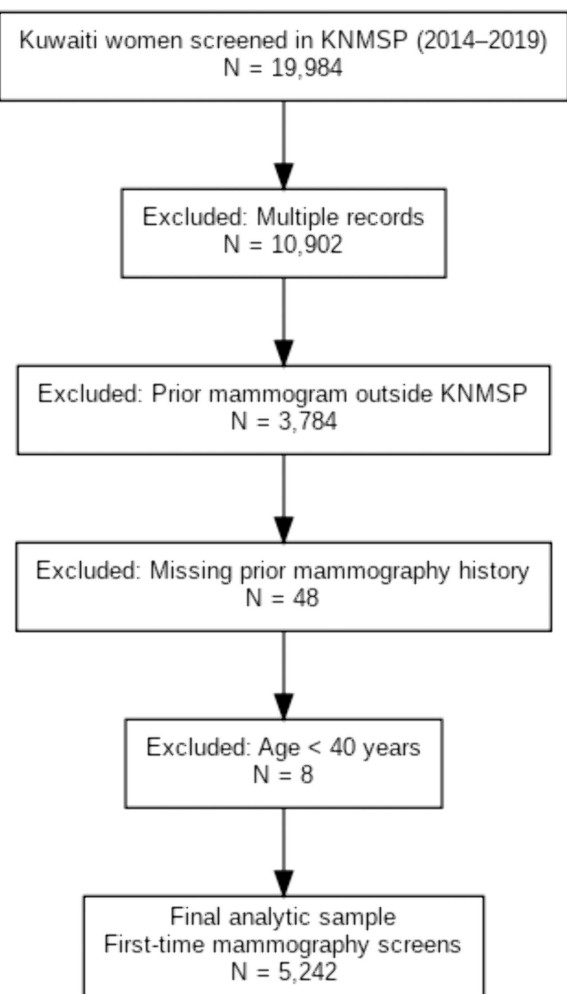

**Fig 1. Flow of participant inclusion and exclusion within the Kuwait National Mammography Screening Program, April 1, 2014–March 31, 2019.**

technicians. All staff underwent standardized training in data collection and entry protocols, and periodic audits of the electronic database were performed by program supervisors to ensure accuracy and consistency. The dataset, covering screenings conducted between April 1, 2014, and March 31, 2019, was provided in a password-protected Microsoft Excel file. Data cleaning procedures were implemented to ensure accuracy and consistency. Implausible height values (<120 cm or >200 cm) were treated as missing. When height and weight values appeared to be transposed, corrections were applied using logical rules (e.g., if reported height < weight), and these were cross-validated with other available anthropometric fields when possible. Values for age at menarche <9 years or menopausal age < 20 years were coded as missing, based on established data standards [11,12]. We applied range checks and logical cross-validations; outliers were set to missing when implausible. Breastfeeding duration categories (never/< 1 month, 1–3, 3–6, 6–12, 12–23, and ≥24 months) were based on both clinical relevance (short vs. prolonged breastfeeding intervals) and the observed distribution in the dataset. Menopausal age categories (premature <40 years, early 40–44, average 45–51, late >51) followed standard clinical definitions, consistent with prior epidemiological literature.

For unmarried women with no reported pregnancies, parity was recorded as "no children," and breastfeeding duration was coded as "never – <1 month." Records missing examination dates were excluded. Duplicate entries were reviewed, and the most complete record for each individual was retained. A variable indicating the number of screenings per participant was created to help verify first-time screenings. Variables with a high proportion of missing data, specifically dietary habits and physical activity, were excluded from the analysis as missingness exceeded 35%. The data in this study were accessed and collected during the period from February 2023 to May 2023.

## Statistical analysis

Categorical variables were compared using Pearson $\chi^2$ tests (association). Monotonic trends across the five ordered screening years were assessed using Cuzick's nonparametric trend test and a logistic model with exam year entered as a continuous predictor. The binary outcome was young (40–49 years = 1) versus 50 + years = 0. We report adjusted ORs and 95% CIs. Multicollinearity was assessed using the variance inflation factor (VIF). Model fit was evaluated with the Hosmer–Lemeshow test. Analyses used Stata 17 [13].

## Ethical considerations

The study adhered to ethical guidelines and did not require access to personal medical records from the Ministry of Health. Data were retrieved from the KNMSP database, anonymized and provided as a password-protected Excel file. Ethical approval was obtained from the Health Sciences Center's Ethics Committee at Kuwait University and the Ministry of Health's Ethical Committee.

## Results

A total of 5,242 records for asymptomatic Kuwaiti women who underwent their first mammography screening were included in the study, of whom 65.0% (n = 3,407) were screened at a younger age (40–49 years) and 35.0% (n = 1,835) at an older age (50 + years). The majority of women screened at a younger age lived in the Capital governorate (29.7%), were university-educated or postgraduates (43.0%), married (94.5%), and underwent screening during the fifth year of the KNMSP (30.7%). Most women screened at a younger age had not performed breast self-examination (71.3%) or clinical breast examination (87.0%). Exam year was associated with screening age group (Pearson $\chi^2$, p < 0.001). Monotonic trend tests likewise indicated increasing odds of screening at 40–49 over time (Cuzick z = 4.56, p < 0.001; per-year logistic OR = 1.10, 95% CI 1.06–1.14).

Approximately 60% of participants reported no prior awareness of mammography screening. Awareness was higher among women screened at a young age compared with 50 + years (39.8% vs. 34.7%, p < 0.001). The source of knowledge about screening also differed significantly: a higher proportion of younger women cited brochures or breast cancer

campaigns (38.6% vs. 32.3%), whereas fewer younger women cited physicians as their primary source (28.5% vs. 38.5%). Given that only 1.8% of participants cited social media, the corresponding estimates should be interpreted with caution. Contraceptive pill use was also significantly higher among young women (40.2% vs. 36.9%, p = 0.023), Table 1.

Most women screened at a younger age had no family history of breast cancer (74.1%) or other cancers (60.5%). Diabetes prevalence was significantly lower in the younger group (7.5% vs. 33.4%, $p < 0.001$). Nearly half of the study population were obese (52.1%), with obesity more prevalent among women screened at an older age (60.6% vs. 47.5%, $p < 0.001$) (Table 2).

Although most women did not use hormone replacement therapy (HRT), use was significantly higher among those screened at a younger age (9.4% vs. 5.2%, $p < 0.001$). Breast surgery was more common in younger-screened women (9.3% vs. 6.3%), whereas ovary removal (3.1% vs. 1.3%) and hysterectomy (5.2% vs. 2.0%) were more common in the older-screened group ($p < 0.001$ for both).

In terms of reproductive history, 3–5 children was the most commonly reported pregnancy frequency among younger women (46.3% vs. 32.2%, $p < 0.001$). In the younger group, 76.1% reported first pregnancy between 20–40 years (20–30: 70.1%; 31–40: 6.1%). Breastfeeding duration of 1–3 months was the most frequently reported in both age groups.

Regarding menopause, most women (56.3%) experienced menopause at the average age of 45–51 years. However, premature and early menopause were more common among menopausal women, premature and early menopause were more common in the young group (15% vs. 3% and 28% vs. 7%, respectively). Menstruating women made up 65.7% of the sample, with a significantly higher proportion in the young group (90.5% vs. 19.5%, $p < 0.001$). Menopausal status was based on self-report of cessation of menstruation, which may overlap with the younger age group.

Additional predictors of younger screening included undergoing clinical breast examination (OR = 1.4, 95% CI: 1.1–1.7, $p = 0.004$), using HRT (OR = 1.7, 95% CI: 1.3–2.2, $p < 0.001$), obtaining screening information from social media (OR = 2.6, 95% CI: 1.3–5.2, $p = 0.008$), and having 1–2, 3–5, or 6–8 children compared to none (ORs: 2.4, 2.7, and 1.9, respectively; $p < 0.001$).

Conversely, being overweight or obese was associated with reduced odds of screening at a younger age compared to having a normal Body Mass Index (BMI) (OR = 0.67–0.53; $p < 0.001$). Hysterectomy was also associated with decreased odds (OR = 0.51, 95% CI: 0.34–0.76, $p = 0.001$).

Breastfeeding duration was inversely associated with screening at a younger age: compared to women who never breastfed or did so for less than one month, those who breastfed for 3–6, 6–12, 12–23, or 24 + months had significantly lower odds of early screening (OR range: 0.41–0.69, $p < 0.05$). No variables were excluded due to multicollinearity (all VIF values < 2). Exploratory interaction terms (e.g., education × awareness, parity × age) were tested but were not statistically significant and were not retained in the final model (Table 3).

As shown in Fig 2, a higher proportion of women screened at a young age reported regular exercise (36% vs. 30%, $p < 0.001$). No significant differences were observed between the groups in terms of fatty food consumption, alcohol intake, or smoking behavior. Statistically significant association ($p < 0.001$) observed only for exercise; p-values were calculated using the chi-square test. These behavioral variables were not retained in the multivariate model because they were either non-significant or subject to high missingness (e.g., dietary data).

## Discussion

This study identifies characteristics associated with undergoing a first mammogram at 40–49 years versus ≥50 years in Kuwait, an area with limited prior evidence both locally and globally. In Kuwait, only 7.8% of the target population underwent mammography screening during the first five years of the national cancer screening program [7]. Therefore, our findings may support increased uptake of screening in Kuwait through creating more targeted interventions and strategies to influence screening and enable improved engagement in the screening programs.

Among all Kuwaiti women screened for the first time at the KNMSP, the majority (65%) were screened at a younger age. Similarly, a study in Korea showed that women aged 65 and above were less likely to undergo mammography

**Table 1. Sociodemographic, Screening Knowledge, and Behavioral Characteristics Associated with Age at First Mammography Screening Among Kuwaiti Women (2014–2019).**

| Variables | Total | Age at first screening | | | |
|---|---|---|---|---|---|
| | | Younger | Older | | |
| | n (%) | (40–49 years) | (50+ years) | | |
| **Total** | **5,242** (100%) | 3,407 (**65.0%**) | 1,835 (**35.0%**) | | |
| **Governorate** | | | | **<0.001** | *** |
| The Capital | 1,557 (29.7%) | 1,010 (64.9%) | 547 (35.1%) | | |
| Ahmadi | 560 (10.7%) | 391 (69.8%) | 169 (30.2%) | | |
| Jahra | 532 (10.1%) | 383 (72.0%) | 149 (28.0%) | | |
| Farwaniya | 790 (15.1%) | 515 (65.2%) | 275 (34.8%) | | |
| Hawalli | 1,339 (25.5%) | 865 (64.6%) | 474 (35.4%) | | |
| Mubarak Al-Kabeer | 455 (8.6%) | 239 (52.5%) | 216 (47.5%) | | |
| **Education** | | | | **<0.001** | *** |
| Illiterate-secondary | 1,138 (21.7%) | 440 (38.7%) | 698 (61.3%) | | |
| High School | 887 (17.0%) | 569 (64.1%) | 318 (35.9%) | | |
| Diploma | 1,310 (25.1%) | 928 (70.8%) | 382 (29.2%) | | |
| University & Postgraduate | 1,888 (36.2%) | 1,459 (77.3%) | 429 (22.7%) | | |
| **Marital status** | | | | 0.320 | |
| Yes | 4,934 (94.2%) | 3,216 (65.2%) | 1,718 (34.8%) | | |
| No | 303 (5.8%) | 189 (62.4%) | 114 (37.6%) | | |
| **The Year of Undergoing Mammography Screening** | | | | **<0.001** | *** |
| Year 1 [Apr,14-Mar,15] | 728 (13.9%) | 459 (63.0%) | 269 (37.0%) | | |
| Year 2 [Apr,15-Mar,16] | 1,064 (20.3%) | 656 (61.7%) | 408 (38.3%) | | |
| Year 3 [Apr,16-Mar,17] | 1,048 (19.9%) | 655 (62.5%) | 393 (37.5%) | | |
| Year 4 [Apr,17-Mar,18] | 920 (17.6%) | 591 (64.2%) | 329 (35.8%) | | |
| Year 5 [Apr,18-Mar,19] | 1,482 (28.3%) | 1,046 (70.6%) | 436 (29.4%) | | |
| **Breast self-examination** | | | | **<0.001** | *** |
| Yes | 1,391 (26.7%) | 958 (68.9%) | 433 (31.1%) | | |
| No | 3,815 (73.3%) | 2,427 (63.6%) | 1,388 (36.4%) | | |
| **Clinical breast examination** | | | | 0.009 | ** |
| Yes | 611 (11.7%) | 426 (69.7%) | 185 (30.3%) | | |
| No | 4,605 (88.3%) | 2,963 (64.3%) | 1,642 (35.7%) | | |
| **Awareness of Screening** | | | | **<0.001** | *** |
| Yes | 1,992 (39.6%) | 1,356 (68.1%) | 636 (31.9%) | | |
| No | 3,040 (60.4%) | 1,909 (62.8%) | 1,131 (37.2%) | | |
| **The Source of Knowledge Regarding Screening** | | | | **<0.001** | *** |
| Brochure & Campaign | 1,905 (36.4%) | 1,314 (69.0%) | 591 (31.0%) | | |
| TV & Newspaper | 545 (10.4%) | 373 (68.4%) | 172 (31.6%) | | |
| Social Media | 94 (1.8%) | 82 (87.2%) | 12 (12.8%) | | |
| Relative or Friend | 900 (17.2%) | 581 (64.6%) | 319 (35.4%) | | |
| Physician | 1,675 (32.0%) | 970 (57.9%) | 705 (42.1%) | | |
| Others | 113 (2.2%) | 82 (72.6%) | 31 (27.4%) | | |
| **Contraceptive Pill Consumption** | | | | 0.023 | * |
| Yes | 2,003 (39.0%) | 1,341 (66.9%) | 662 (33.1%) | | |
| No | 3,129 (61.0%) | 1,998 (63.9%) | 1,131 (36.1%) | | |
| **Duration of Consuming Contraceptive Pill** | | | | 0.016 | * |
| [0–5] months | 775 (39.0%) | 542 (69.9%) | 233 (30.1%) | | |

*(Continued)*

**Table 1.** (Continued)

| Variables | Total | Age at first screening | | | |
|---|---|---|---|---|---|
| | | Younger | Older | | |
| | n (%) | (40–49 years) | (50+years) | | |
| [6–10] months | 575 (28.9%) | 385 (67.0%) | 190 (33.0%) | | |
| [11–19] months | 363 (18.2%) | 240 (66.1%) | 123 (33.9%) | | |
| 20+months | 273 (13.7%) | 162 (59.3%) | 111 (40.7%) | | |

χ² test: *p<0.05, **p**<0.01, ***p***<0.001. Totals=column %, Younger/Older=n (row %). Percentages based on non-missing data.

compared to those aged 40–49 [14]. In contrast, a UAE study found that younger women (40–49) were less likely to screen than those aged 50 and older, as they believed screening was unnecessary before 50 [15]. A U.S. study also reported lower screening rates among women aged 40–49, attributing it to lower education and financial barriers [16]. These discrepancies likely reflect varying social, economic, and health system factors across countries, emphasizing the need for localized screening strategies.

Although our analysis focused on predictors of earlier first screening, understanding delays among women aged 50+ is equally important. Potential explanations include competing comorbidities, lower perceived benefit before a physician's recommendation, or practical barriers such as transport and appointment scheduling. Future qualitative work with older women could clarify these issues and guide age-specific interventions to reduce delayed screening.

More than half of the women in our study were unaware of breast cancer screening. However, younger women were more aware than older women. This aligns with a UAE study showing higher knowledge scores among younger, more educated women [15]. Similar findings were reported in Ghana and the U.S., where increased knowledge was associated with greater screening uptake [17,18]. These findings highlight the critical need to improve screening awareness through tailored campaigns that address the motivational factors influencing participation by age group.

The source of screening knowledge differed by age: younger women were more influenced by brochures and campaigns, while older women relied more on physicians. Women citing social media as their main source were more likely to screen earlier (adjusted OR ≈ 2.6); however, this subgroup was small (n=94; 82 in 40–49 vs. 12 in 50+), and the wide confidence interval indicates imprecision which should be interpreted cautiously. This likely reflects both KNMSP-linked campaigns and self-selection by health-motivated younger women. In contrast, physician- or peer-informed women were less likely to screen early, consistent with regional studies from Jordan and the UAE showing the importance of physician recommendation [19,20] and with Ethiopian findings that physicians were the dominant influence [21]. Together, these observations suggest that social media could be a promising but still underutilized tool for promoting earlier screening in Kuwait. Additionally, reverse causation cannot be excluded (e.g., women already motivated to screen may also be more engaged online), underscoring the need for prospective evaluations of digital outreach.

Higher educational attainment was significantly associated with screening at a younger age. Women with a university or postgraduate degree had nearly five times greater odds of early screening compared to those with no or low education. Similar associations have been reported in the UAE, Saudi Arabia, Ghana, and Greece [20,22–24]. These findings reinforce the need to enhance outreach to less educated women through simplified, accessible, and culturally tailored messaging.

Our results showed that most women had not performed clinical or self-breast examinations. However, clinical examination was significantly associated with early mammography screening. This suggests that healthcare providers may influence women's screening decisions through clinical contact, as supported by UAE studies [20]. Encouraging women to begin with clinical evaluations may encourage subsequent adherence to mammography screening, though prospective evaluation is warranted.

**Table 2. Health Status, Reproductive History, and Maternal Health Factors Associated with Age at First Mammography Screening Among Kuwaiti Women (2014–2019).**

| Variables | Total | Age at first screening Younger | Older | | |
|---|---|---|---|---|---|
| | **n (%)** | **(40–49 years)** | **(50 + years)** | **p-value** | |
| **Total** | **5,242** (100%) | 3,407 (**65.0%**) | 1,835 (**35.0%**) | | |
| **History Of Prior Cancer** | | | | 0.970 | |
| Yes | 46 (0.9%) | 30 (65.2%) | 16 (34.8%) | | |
| No | 5,173 (99.1%) | 3,360 (65.0%) | 1,813 (35.0%) | | |
| **Family History of Breast Cancer** | | | | **0.003** | ** |
| Yes | 1,287 (24.6%) | 881 (68.5%) | 406 (31.5%) | | |
| No | 3,938 (75.4%) | 2,516 (63.9%) | 1,422 (36.1%) | | |
| **Family History of Other Cancers** | | | | **<0.001** | *** |
| Yes | 1,970 (37.9%) | 1,333 (67.7%) | 637 (32.3%) | | |
| No | 3,232 (62.1%) | 2,045 (63.3%) | 1,187 (36.7%) | | |
| **Diabetes** | | | | **<0.001** | *** |
| Yes | 339 (15.6%) | 111 (32.7%) | 228 (67.3%) | | |
| No | 1,829 (84.4%) | 1,375 (75.2%) | 454 (24.8%) | | |
| **Body Mass Index (BMI)** | | | | **<0.001** | *** |
| Normal | 675 (12.9%) | 515 (76.3%) | 160 (23.7%) | | |
| Overweight | 1,816 (34.9%) | 1,264 (69.6%) | 552 (30.4%) | | |
| Obese | 2,705 (52.1%) | 1,610 (59.5%) | 1,095 (40.5%) | | |
| **Hormone Replacement Therapy** | | | | **<0.001** | *** |
| Yes | 412 (7.9%) | 318 (77.2%) | 94 (22.8%) | | |
| No | 4,779 (92.1%) | 3,055 (63.9%) | 1,724 (36.1%) | | |
| **Duration Of Hormone Replacement Therapy** | | | | **0.018** | * |
| [1–2] years | 249 (62.8%) | 199 (79.9%) | 50 (20.1%) | | |
| [3–5] years | 71 (17.9%) | 57 (80.3%) | 14 (19.7%) | | |
| [6–10] years | 48 (12.1%) | 29 (60.4%) | 19 (39.6%) | | |
| [11–19] years | 22 (5.5%) | 18 (81.8%) | 4 (18.2%) | | |
| 20 + years | 6 (1.5%) | 3 (50.0%) | 3 (50.0%) | | |
| **Underwent A Breast Surgery** | | | | **<0.001** | *** |
| Yes | 432 (8.3%) | 316 (73.1%) | 116 (26.9%) | | |
| No | 4,796 (91.7%) | 3,080 (64.2%) | 1,716 (35.8%) | | |
| **Radiation** | | | | 0.707 | |
| Yes | 10 (0.2%) | 6 (60.0%) | 4 (40.0%) | | |
| No | 4,437 (99.8%) | 2,913 (65.7%) | 1,524 (34.3%) | | |
| **Ovary Removal** | | | | **<0.001** | *** |
| Yes | 99 (1.9%) | 43 (43.4%) | 56 (56.6%) | | |
| No | 5,108 (98.1%) | 3,345 (65.5%) | 1,763 (34.5%) | | |
| **Hysterectomy** | | | | **<0.001** | *** |
| Yes | 163 (3.1%) | 69 (42.3%) | 94 (57.7%) | | |
| No | 5,041 (96.9%) | 3,316 (65.8%) | 1,725 (34.2%) | | |
| **Age at Menarche** | | | | 0.275 | |
| <12 years old | 645 (12.5%) | 434 (67.3%) | 211 (32.7%) | | |
| 12-14 years old | 3,958 (76.8%) | 2,579 (65.2%) | 1,379 (34.8%) | | |
| 15 years old and above | 549 (10.6%) | 345 (62.8%) | 204 (37.2%) | | |
| **Frequency of Pregnancy** | | | | **<0.001** | *** |

*(Continued)*

**Table 2.** (Continued)

| Variables | Total | Age at first screening | | | |
|---|---|---|---|---|---|
| | | Younger | Older | | |
| | n (%) | (40–49 years) | (50+years) | p-value | |
| No child | 382 (7.3%) | 228 (59.7%) | 154 (40.3%) | | |
| [1–2] children | 456 (8.7%) | 316 (69.3%) | 140 (30.7%) | | |
| [3–5] children | 2,172 (41.8%) | 1,575 (72.5%) | 597 (27.5%) | | |
| [6–8] children | 1,718 (33.1%) | 1,065 (62.0%) | 653 (38.0%) | | |
| [9–16] children | 463 (8.9%) | 192 (41.5%) | 271 (58.5%) | | |
| **Age at First Pregnancy** | | | | **<0.001** | *** |
| <20 years old | 1,362 (28.3%) | 743 (54.6%) | 619 (45.4%) | | |
| 20-30 years old | 3,162 (65.7%) | 2,205 (69.7%) | 957 (30.3%) | | |
| 31-40 years old | 270 (5.6%) | 191 (70.7%) | 79 (29.3%) | | |
| 40+year old | 13 (0.2%) | 7 (53.8%) | 6 (46.2%) | | |
| **Breastfeeding History** | | | | 0.419 | |
| Yes | 4,397 (87.9%) | 2,853 (64.9%) | 1,544 (35.1%) | | |
| No | 604 (12.1%) | 402 (66.6%) | 202 (33.4%) | | |
| **Duration of Breastfeeding** | | | | **<0.001** | *** |
| Never to < 1 month | 755 (15.7%) | 569 (75.4%) | 186 (24.6%) | | |
| [1–3] months | 1,504 (31.3%) | 1,075 (71.5%) | 429 (28.5%) | | |
| [3–6] months | 873 (18.2%) | 593 (67.9%) | 280 (32.1%) | | |
| [6–12] months | 725 (15.1%) | 431 (59.4%) | 294 (40.6%) | | |
| [12–23] months | 160 (3.3%) | 93 (58.1%) | 67 (41.9%) | | |
| 24+months | 793 (16.5%) | 392 (49.4%) | 401 (50.6%) | | |
| **Age at Menopause** | | | | **<0.001** | *** |
| Premature (<40 y) | 93 (5.3%) | 45 (48.4%) | 48 (51.6%) | | |
| Early (40–44 y) | 188 (10.8%) | 84 (44.7%) | 104 (55.3%) | | |
| Average (45-51y) | 979 (56.4%) | 171 (17.5%) | 808 (82.5%) | | |
| Late (>51 y) | 475 (27.3%) | 0 (0.0%) | 475 (100.0%) | | |
| **Menstruation status** | | | | **<0.001** | *** |
| Yes | 3,439 (65.7%) | 3,082 (89.6%) | 357 (10.4%) | | |
| No | 1,788 (34.2%) | 314 (17.6%) | 1,474 (82.4%) | | |

χ² test: *p*< 0.05, **p**< 0.01, ***p***< 0.001. Totals = column %, Younger/Older = n (row %). Percentages based on non-missing data.

A significant association was observed between a family history of breast cancer and screening at a younger age. Other studies have reported similar findings [25,26], although some suggest that fear of cancer may deter screening [27]. Clinicians should recognize and address both risk awareness and fear when advising patients on screening. Being overweight or obese was associated with lower odds of early screening. This supports findings from the U.S. [16], although other studies, such as among African-American women, found increased screening in obese populations [28]. This discrepancy could reflect differing health behaviors and cultural perceptions of body weight, highlighting the potential value of targeted messaging for overweight and obese women. Contrary to several Western studies suggesting that higher parity reduced screening adherence [29–31], our findings showed that pregnancy frequency was positively associated with early screening. This was supported by a Qatari study [32], suggesting that tailoring screening strategies to regional reproductive norms may be important. We also found that longer breastfeeding duration (3–24+months) was associated with lower odds of screening at a younger age. This agrees with Australian findings [25], although an Iranian study reported no

**Table 3. Multivariable logistic regression of characteristics associated with screening at 40–49 Years (vs. ≥ 50), Kuwait (2014–2019).**

| Characteristics | First screen at a younger age | | | |
|---|---|---|---|---|
| | OR | (95% CI) | *p*-value | |
| **Governorate** | | | | |
| Capital | Reference group | | | |
| Al-Ahmadi | 2.489 | (1.91-3.24) | **<0.001** | *** |
| Al-Jahra | 2.614 | (1.98-3.45) | **<0.001** | *** |
| Farwaniya | 1.395 | (1.11-1.74) | **0.004** | ** |
| Hawalli | 1.138 | (0.95-1.37) | 0.165 | |
| Mubarak Al-Kabeer | 0.885 | (0.68-1.15) | 0.355 | |
| **Body Mass Index (BMI)** | | | | |
| Normal | Reference group | | | |
| Overweight | 0.670 | (0.53-0.85) | **0.001** | ** |
| Obese | 0.525 | (0.42-0.66) | **<0.001** | *** |
| **Education** | | | | |
| Illiterate-secondary | Reference group | | | |
| High School | 2.275 | (1.83-2.83) | **<0.001** | *** |
| Diploma | 3.087 | (2.52-3.79) | **<0.001** | *** |
| University & Postgraduate | 4.530 | (3.71-5.53) | **<0.001** | *** |
| **Clinical breast examination** | | | | |
| No | Reference group | | | |
| Yes | 1.391 | (1.11-1.74) | **0.004** | ** |
| **The Source of Knowledge Regarding Screening** | | | | |
| Brochure & Campaign | Reference group | | | |
| TV & Newspaper | 0.918 | (0.71-1.19) | 0.518 | |
| Social Media | 2.576 | (1.28-5.19) | **0.008** | ** |
| Relative or Friend | 0.709 | (0.58-0.87) | **0.001** | ** |
| Physician | 0.692 | (0.59-0.82) | **<0.001** | *** |
| Others | 1.156 | (0.69-1.94) | 0.583 | |
| **Frequency of Pregnancy** | | | | |
| No child | Reference group | | | |
| [1-2] children | 2.420 | (1.59-3.69) | **<0.001** | *** |
| [3-5] children | 2.657 | (1.85-3.82) | **<0.001** | *** |
| [6-8] children | 1.889 | (1.30-2.74) | **0.001** | ** |
| [9-16] children | 1.130 | (0.74-1.73) | 0.570 | |
| **Duration of Breastfeeding** | | | | |
| never to less than 1 month | Reference group | | | |
| [1-3] months | 0.786 | (0.58-1.06) | 0.116 | |
| [3-6] months | 0.685 | (0.50-0.94) | **0.019** | * |
| [6-12] months | 0.505 | (0.37-0.70) | **<0.001** | *** |
| [12-23] months | 0.494 | (0.32-0.78) | **0.002** | ** |
| 24+ months | 0.414 | (0.30-0.57) | **<0.001** | *** |
| **Hormone Replacement Therapy** | | | | |
| No | Reference group | | | |
| Yes | 1.688 | (1.28-2.23) | **<0.001** | *** |
| **Hysterectomy** | | | | |
| No | Reference group | | | |
| Yes | 0.507 | (0.34-0.76) | **0.001** | ** |

Hosmer–Lemeshow goodness-of-fit test: p = 0.834 (model fits well). *p < 0.05, **p < 0.01, ***p < 0.001. Bold denotes statistical significance.

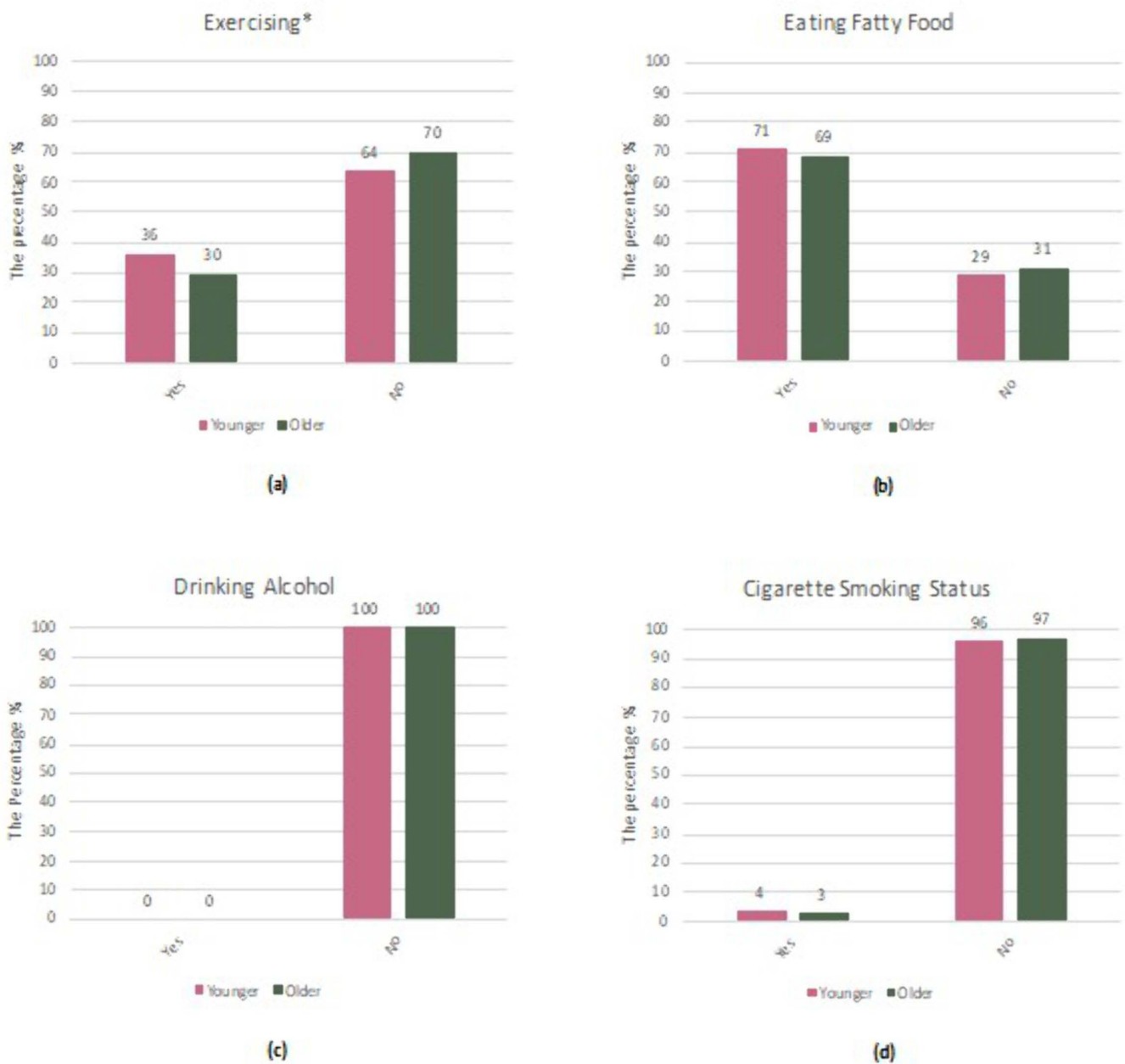

**Fig 2. Proportion of selected behavioral factors among women undergoing their first mammography screening at a younger (40–49 years) vs. older age (50+), Kuwait, 1 April 2014 – 31 March 2019.** Subfigures: (a) regular exercise, (b) fatty food consumption, (c) alcohol use, (d) cigarette smoking status.

significant association [26]. The Iranian study did not examine breastfeeding duration, which may explain the difference. Further research is needed to clarify how breastfeeding behaviors influence screening timing. Use of hormone replacement therapy (HRT) was associated with early screening, consistent with studies from Australia and elsewhere [25,33,34]. However, other studies found no association [31], suggesting that contextual or cultural factors may moderate this

relationship. In our setting, HRT use may proxy increased healthcare contact during menopause care rather than being a causal driver of earlier screening; embedding screening prompts in menopause clinics could leverage this touchpoint.

## Strengths and limitations

Although the dataset covers about 8% of the target population, it comes from Kuwait's first and only structured, population-based breast cancer screening program and provides early insights into screening behaviors. Analyses were restricted to Kuwaiti nationals because KNMSP serves only Kuwaiti women; non-Kuwaiti residents were largely screened outside KNMSP, often in the private sector, and those appearing in KNMSP were typically high-risk diagnostic referrals, which limits generalizability to non-Kuwaiti residents. Private-sector mammography is not centrally recorded, so screening outside KNMSP could not be quantified; participation estimates and inferences therefore pertain to Kuwaiti women screened within KNMSP. The cohort comprised asymptomatic first-time KNMSP screens, which reduces but does not eliminate selection bias. Some variables had substantial missingness, particularly those introduced in later program years, and were excluded.

## Conclusion

This study provides valuable insights into the characteristics associated with younger women (aged 40–49) undergoing their first mammography screening in Kuwait. Key factors such as higher education, increased pregnancy frequency, undergoing clinical breast examination, using hormone replacement therapy, and having a normal BMI were significantly associated with screening at a younger age. By identifying these sociodemographic, behavioral, and health-related factors, our findings can guide the development of more targeted and culturally tailored breast cancer screening initiatives. Targeted, culturally tailored strategies, particularly for women with lower education or higher BMI, may enhance participation and support earlier detection. Longitudinal evaluation is needed to determine whether earlier first screening translates into down-staging and improved outcomes.

## Author contributions

**Conceptualization:** Eiman Alawadhi, Alyah A. Al-Yahya, Batool A. Mohammed, Mariam M. Al-Ruwaie, Sarah H. Al-Mutairi.

**Data curation:** Eiman Alawadhi, Alyah A. Al-Yahya, Batool A. Mohammed, Mariam M. Al-Ruwaie, Sarah H. Al-Mutairi.

**Formal analysis:** Eiman Alawadhi, Alyah A. Al-Yahya, Batool A. Mohammed, Mariam M. Al-Ruwaie, Sarah H. Al-Mutairi.

**Investigation:** Eiman Alawadhi, Alyah A. Al-Yahya, Batool A. Mohammed, Mariam M. Al-Ruwaie, Sarah H. Al-Mutairi.

**Methodology:** Eiman Alawadhi, Alyah A. Al-Yahya, Batool A. Mohammed, Mariam M. Al-Ruwaie, Sarah H. Al-Mutairi.

**Resources:** Alyah A. Al-Yahya, Batool A. Mohammed, Mariam M. Al-Ruwaie, Sarah H. Al-Mutairi.

**Software:** Eiman Alawadhi, Alyah A. Al-Yahya, Batool A. Mohammed, Mariam M. Al-Ruwaie, Sarah H. Al-Mutairi.

**Supervision:** Eiman Alawadhi.

**Validation:** Eiman Alawadhi, Alyah A. Al-Yahya, Batool A. Mohammed, Mariam M. Al-Ruwaie, Sarah H. Al-Mutairi.

**Writing – original draft:** Eiman Alawadhi.

**Writing – review & editing:** Alyah A. Al-Yahya, Batool A. Mohammed, Mariam M. Al-Ruwaie, Sarah H. Al-Mutairi, Nourah Alsheridah.

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
