## [Decision Letter · Decision Letter 0]

27 Aug 2025

PGPH-D-25-01572

Characteristics associated with women undergoing their first mammography screening at a younger age

Dear Dr. Alsheridah,

Thank you for submitting your manuscript to PLOS Global Public Health. After careful consideration, we feel that it has merit but does not fully meet PLOS Global Public Health’s publication criteria as it currently stands. Therefore, we invite you to submit a revised version of the manuscript that addresses the points raised during the review process.

The reviewer comments are detailed in an attached document. Please ensure you open this and address these comments.

We look forward to receiving your revised manuscript.

Kind regards,

Daniel Parkes, PhD

Staff Editor

Journal Requirements:

1. In the online submission form, you indicated that “The data supporting the findings of this study are protected and available upon reasonable request from Kuwait Ministry of Health.”

a) In a public repository, 

b) Within the manuscript itself, or 

c) Uploaded as supplementary information.

This policy applies to all data except where public deposition would breach compliance with the protocol approved by your research ethics board. If your data cannot be made publicly available for ethical or legal reasons (e.g., public availability would compromise patient privacy), please explain your reasons by return email and your exemption request will be escalated to the editor for approval. Your exemption request will be handled independently and will not hold up the peer review process, but will need to be resolved should your manuscript be accepted for publication. One of the Editorial team will then be in touch if there are any issues."

2. Please provide separate main figure files in .tif or .eps format only and ensure that all files are under our size limit of 10MB.

Additional Editor Comments (if provided):

Reviewers' comments:

Reviewer's Responses to Questions

**Comments to the Author**

1. Does this manuscript meet PLOS Global Public Health’s publication criteria ? Is the manuscript technically sound, and do the data support the conclusions? The manuscript must describe methodologically and ethically rigorous research with conclusions that are appropriately drawn based on the data presented.

Reviewer #1: Yes

2. Has the statistical analysis been performed appropriately and rigorously?

Reviewer #1: Yes

3. Have the authors made all data underlying the findings in their manuscript fully available (please refer to the Data Availability Statement at the start of the manuscript PDF file)?

Reviewer #1: Yes

4. Is the manuscript presented in an intelligible fashion and written in standard English?

Reviewer #1: Yes

5. Review Comments to the Author

Reviewer #1: Thank you for the opportunity to review this important and well-written manuscript. The study addresses a critical gap in the literature by examining the characteristics associated with age at first mammography screening among Kuwaiti women using data from a national screening program. The focus on younger women (40–49 years) is timely given recent updates to screening guidelines and low participation rates in the region. The manuscript is generally well-structured and clearly presented, with thoughtful interpretations.

6. PLOS authors have the option to publish the peer review history of their article (what does this mean? ). If published, this will include your full peer review and any attached files.

**Do you want your identity to be public for this peer review?** For information about this choice, including consent withdrawal, please see our Privacy Policy .

Reviewer #1: No

---

## [Decision Letter · Decision Letter 1]

10 Oct 2025

PGPH-D-25-01572R1

Characteristics associated with women undergoing their first mammography screening at a younger age

Dear Dr. Nourah Alsheridah

Thank you for submitting your manuscript to PLOS Global Public Health. After careful consideration, we feel that it has merit but does not fully meet PLOS Global Public Health’s publication criteria as it currently stands. Therefore, we invite you to submit a revised version of the manuscript that addresses the points raised during the review process.

We look forward to receiving your revised manuscript.

Kind regards,

Titilayo Abike Olaoye, PhD

Academic Editor

Journal Requirements:

Additional Editor Comments:

Overall Assessment

This is a well-organized, methodologically sound, and clearly written article that makes a valuable contribution to understanding determinants of early mammography screening among Kuwaiti women. The manuscript aligns well with the aims of PLOS Global Public Health and effectively contextualizes findings within regional and global literature. The study design, analysis, and discussion are appropriate and transparent. However, a few minor revisions can enhance clarity, flow, and consistency.

Strengths

Relevance: Addresses a timely and important public health issue (low mammography uptake and early screening predictors).

Data Quality: Uses a national dataset (KNMSP) with clear inclusion/exclusion criteria.

Methodological Rigor: Clear regression modeling, checks for collinearity, and use of Cuzick trend tests.

Interpretation: Thoughtful comparison with international literature.

Compliance: Ethical approvals, data availability, and funding declarations all meet journal standards.

Specific Comments

1. Abstract: The abstract is comprehensive but can be slightly condensed for clarity

Line 61–63: “Targeted, culturally tailored outreach … may promote earlier participation.” → Consider rephrasing to “Targeted, culturally sensitive outreach, particularly using digital media, may encourage earlier participation.”

Consider limiting numerical details (e.g., reduce the number of odds ratios) to keep the focus on the most salient findings.

2. Introduction: Clarify the rationale for focusing on first-time screening at younger age — one line linking this to prevention and down-staging would strengthen justification (after line 98).

Line 91–94: The phrase “7.8% screened via KNMSP in its first five years vs a 70% target” could be supported with a citation to emphasize this contrast.

3. Methods: Line 111–113: “To assess factors associated with screening at 40–49 vs 50+ years” – specify that this outcome was binary (younger vs older age) for clarity.

Data Cleaning Section: Consider moving the range-check explanations (lines 137–146) to supplementary materials to streamline the main text.

4. Results

Tables are well structured. Ensure uniform alignment and spacing across tables (some inconsistent tab spacing observed). Consider referencing Table 3 explicitly in the text where regression findings are summarized (line 210–223) to guide the reader.

Figure 2: The captions are informative but could briefly summarize key findings (e.g., “Younger women reported more exercise (p<0.001) but no difference in diet or smoking”).

5. Discussion: Lines 252–256: Consider adding a brief interpretive statement linking low uptake among older women to potential barriers (transportation, health literacy, physician referral gaps).

Lines 265–276: When discussing social media, clarify that the subgroup was small (as already noted) to pre-empt over interpretation.

Line 281: “Simplified, accessible, and culturally tailored messaging” – consider giving one example (e.g., mosque-based outreach or social media campaigns).

Line 309: “Embedding screening prompts in menopause clinics” – a strong applied recommendation; consider moving this to a short “Implications for Practice” subsection or paragraph for emphasis.

6. References: Ensure all journal names are italicized per PLOS style.

Cross-check in-text citation numbers for consistency (especially references 15–20, where multiple UAE studies are cited closely).

7. Formatting: The manuscript occasionally shows leftover line numbers (e.g., “Background 43,” “Results 168”). These should be removed in the final submission file. Confirm consistent use of non-breaking spaces in numeric expressions (e.g., “95% CI 1.06–1.14” not “95 % CI”).

Reviewers' comments:

Reviewer's Responses to Questions

**Comments to the Author**

1. If the authors have adequately addressed your comments raised in a previous round of review and you feel that this manuscript is now acceptable for publication, you may indicate that here to bypass the “Comments to the Author” section, enter your conflict of interest statement in the “Confidential to Editor” section, and submit your "Accept" recommendation.

Reviewer #1: All comments have been addressed

2. Does this manuscript meet PLOS Global Public Health’s publication criteria ? Is the manuscript technically sound, and do the data support the conclusions? The manuscript must describe methodologically and ethically rigorous research with conclusions that are appropriately drawn based on the data presented.

Reviewer #1: Yes

3. Has the statistical analysis been performed appropriately and rigorously?

Reviewer #1: Yes

4. Have the authors made all data underlying the findings in their manuscript fully available (please refer to the Data Availability Statement at the start of the manuscript PDF file)?

Reviewer #1: Yes

5. Is the manuscript presented in an intelligible fashion and written in standard English?

Reviewer #1: Yes

6. Review Comments to the Author

Reviewer #1: The manuscript has been substantially improved in response to the previous round of comments. The authors have adequately addressed the concerns raised, and I have no further comments.

7. PLOS authors have the option to publish the peer review history of their article (what does this mean? ). If published, this will include your full peer review and any attached files.

**Do you want your identity to be public for this peer review?** For information about this choice, including consent withdrawal, please see our Privacy Policy .

Reviewer #1: No

 Figure Resubmissions:

---

## [Editor Report · Decision Letter 2]

27 Oct 2025

Characteristics associated with women undergoing their first mammography screening at a younger age

PGPH-D-25-01572R2

Dear Dr Nourah Alsheriddah

We are pleased to inform you that your manuscript 'Characteristics associated with women undergoing their first mammography screening at a younger age' has been provisionally accepted for publication in PLOS Global Public Health.

Best regards,

Titilayo Abike Olaoye, PhD

Academic Editor